# The Presence of T Allele (rs35705950) of the MUC5B Gene Predicts Lower Baseline Forced Vital Capacity and Its Subsequent Decline in Patients with Hypersensitivity Pneumonitis

**DOI:** 10.3390/ijms241310748

**Published:** 2023-06-28

**Authors:** Katarzyna B. Lewandowska, Monika Szturmowicz, Urszula Lechowicz, Monika Franczuk, Katarzyna Błasińska, Maria Falis, Kamila Błaszczyk, Małgorzata Sobiecka, Dorota Wyrostkiewicz, Izabela Siemion-Szcześniak, Małgorzata Bartosiewicz, Piotr Radwan-Röhrenschef, Adriana Roży, Joanna Chorostowska-Wynimko, Witold Z. Tomkowski

**Affiliations:** 11st Department of Lung Diseases, National Research Institute of Tuberculosis and Lung Diseases, Płocka 26, 01-138 Warsaw, Poland; monika.szturmowicz@gmail.com (M.S.);; 2Department of Genetics and Clinical Immunology, National Research Institute of Tuberculosis and Lung Diseases, Płocka 26, 01-138 Warsaw, Poland; ulka100@gmail.com (U.L.);; 3Department of Respiratory Physiopathology, National Research Institute of Tuberculosis and Lung Diseases, Płocka 26, 01-138 Warsaw, Poland; monika.franczuk@gmail.com; 4Department of Radiology, National Research Institute of Tuberculosis and Lung Diseases, Płocka 26, 01-138 Warsaw, Poland

**Keywords:** MUC5B polymorphism, hypersensitivity pneumonitis, disease progression, response to treatment

## Abstract

Hypersensitivity pneumonitis (HP) is an exposure-related interstitial lung disease with two phenotypes—fibrotic and non-fibrotic. Genetic predisposition is an important factor in the disease pathogenesis and fibrosis development. Several genes are supposed to be associated with the fibrosing cascade in the lungs. One of the best-recognized and most prevalent is the common MUC5B gene promoter region polymorphism variant rs35705950. The aim of our study was to establish the frequency of the minor allele of the MUC5B gene in the population of patients with HP and to find the relationship between the MUC5B promoter region polymorphism and the development of lung fibrosis, the severity of the disease course, and the response to the treatment in patients with HP. Eighty-six consecutive patients with HP were tested for the genetic variant rs35705950 of the MUC-5B gene. Demographic, radiological, and functional parameters were collected. The relationship between the presence of the T allele and lung fibrosis, pulmonary function test parameters, and the treatment response were analyzed. The minor allele frequency in the study group was 17%, with the distribution of the genotypes GG in 69.8% of subjects and GT/TT in 30.2%. Patients with the GT/TT phenotype had significantly lower baseline forced vital capacity (FVC) and significantly more frequently had a decline in FVC with time. The prevalence of lung fibrosis in high-resolution computed tomography (HRCT) was not significantly increased in GT/TT variant carriers compared to GG ones. The patients with the T allele tended to respond worse to immunomodulatory treatment and more frequently received antifibrotic drugs. In conclusions: The frequency of MUC5B polymorphism in HP patients is high. The T allele may indicate a worse disease course, worse immunomodulatory treatment response, and earlier need for antifibrotic treatment.

## 1. Introduction

Hypersensitivity pneumonitis (HP) is an interstitial lung disease (ILD) that develops as a result of exposure to different particulates, usually organic (antigens) but also inorganic (haptens) [1]. According to the recent American Thoracic Society (ATS)/Japanese Respiratory Society (JRS)/Latin American Thoracic Association (ALAT) guidelines, two types of HP are distinguished—fibrotic and non-fibrotic [2]. Non-fibrotic HP may resolve completely after exposure elimination, and, in the majority of patients, it is combined with a good prognosis [3]. Fibrotic HP may progress despite the antigen elimination, and results in progressive pulmonary fibrosis (PPF) in as much as 20% of patients [4].

Genetic predisposition to the development of lung fibrosis has been the subject of extensive scientific research recently. Several genome-wide association studies (GWAS) revealed a number of genes involved in the risk of fibrosis development, i.e., telomerase reverse transcriptase (TERT), CST complex subunit STN1, desmoplakin (DSP), dipeptidyl peptidase (DPP9), Toll interacting protein (TOLLIP), and A-kinase anchoring protein 13 (AKAP13) [5,6,7,8,9,10].

One of the best-recognized and most prevalent genetic disorders is the common MUC5B promoter polymorphism variant rs35705950. The MUC5B gene is located in chromosome 11p15.5 and encodes the high-molecular-weight glycoprotein mucin 5B, which increases the viscosity of mucus gel in the airways. MUC5B is expressed in human airway mucosa, and its product, mucin 5B, is present in the tracheobronchial epithelium [11,12]. Mucin 5B is a key molecule in maintaining lung immune homeostasis and mucociliary clearance; thus, it plays a vital role in controlling infections [13]. The lack of MUC5B in mice resulted in impaired airway clearance and decreased survival due to disseminated infections [13]. MUC5B also influences macrophage accumulation and activity. The absence of mucin 5B results in the increase of macrophages’ apoptosis. No such effect was observed in other types of MUC deficiency [11]. On the other hand, the increase in mucin 5B concentration in distal airways, as a result of gain-of-function rs35705950 polymorphism, produced a more significant fibrotic response to bleomycin in mice [14].

The increased expression of MUC5B leads to the increased production and secretion of viscous mucus, which impairs mucociliary clearance and causes increased epithelial susceptibility to injury [15]. The influence of MUC5B polymorphism on the development and progression of lung fibrosis was extensively investigated in idiopathic pulmonary fibrosis (IPF), as the bronchioloalveolar epithelial injury is the main pathogenic mechanism of progressive lung fibrosis in this disease. Such injury may be caused by cigarette smoking, air pollution, or exposure to different dust components and toxic substances [16]. Nevertheless, the above-mentioned factors are not the only triggers of lung fibrosis in IPF. It is estimated that MUC5B polymorphism (the presence of the T minor allele) increases the risk of lung fibrosis by 30–35%, thus being the most substantial single risk factor of developing IPF. Therefore, MUC5B polymorphism may be considered a marker of preclinical pulmonary fibrosis, enabling the identification of the cohort of people at an increased risk of IPF development [17,18]. 

Genetic susceptibility may also play a role in the development of the other fibrosing ILDs [19].

It is suggested that microscopic honeycombing, the hallmark of fibrosis, develops as the result of the aberrant differentiation of progenitor cells located in the distal airways [15]. These cells reveal an increased expression of MUC5B. As the honeycombing is also diagnosed in chronic fibrotic HP and in other fibrotic ILDs, it is reasonable to speculate that the MUC5B polymorphism may have a pathogenic influence on lung fibrosis development in these conditions [17,18]. 

The data on the role of genetic predisposition to HP development are scarce. It is known that some subjects exposed to organic antigens develop allergic alveolitis, but most do not. Farmer’s lung is linked to the gene polymorphism in the tumor necrosis factor-alpha 308 (TNF-alpha-308) promoter as well as human leucocyte antigen (HLA)-A, B, and C loci, pigeon–breeder’s lung with HLA-DR3 allele, and Japanese summer-type HP with HLA—DQ3 allele [20]. The significance of MUC5B polymorphism in HP patients has not been extensively investigated. Only two publications have addressed this problem so far [21,22]. 

Thus, the aim of the present study was to establish the frequency of the minor allele of the MUC5B gene in the population of HP patients and to find the relationship between the MUC5B promoter region polymorphism and the development of lung fibrosis, the severity of the disease course, and the response to treatment.

## 2. Results

### 2.1. Baseline Characteristics of the Study Group

Genetic testing was performed on 86 consecutive HP patients admitted to a single pulmonary department between 1 April 2019 and 31 December 2022. Thirty-eight of them (44%) were male, the median age in the whole group was 58 years (20–75), and the median follow-up time was 67.2 (±48.73) months. Twenty-eight incident and fifty-eight prevalent cases were included. Incident cases did not differ from prevalent ones regarding baseline characteristics and genetic performance (Appendix A). The minor allele frequency (MAF) was 17.4%. The GG genotype was observed in 60 (69.8%) of patients, GT in 24 (27.9%), and TT in 2 (2.3%). The group of 26 patients with either the GT genotype or TT genotype (GT/TT) was compared with 60 patients presenting the GG genotype. The detailed characteristics of the study group according to the MUC5B genotype are presented in Table 1.

Patients with at least one T allele had significantly lower baseline FVC%pred than those with the GG genotype (72.28% vs. 82.18%, *p* = 0.0315). They also tended to have fibrosis in HRCT more often and were more frequently treated with antifibrotics (nintedanib); nevertheless, the differences were not statistically significant. There were no differences between the GG and GT/TT groups regarding age, sex, follow-up time, smoking status, 6 min walking distance (6MWD), distance–saturation product (DSP) [23], TL,co%pred, or gender–age–physiology (GAP) index [24].

### 2.2. FVC Trajectory in Time

In 82 subjects, at least two measurements of FVC were performed. In that group, the assessment of median annual change in FVC%pred was possible. In the whole group, 42 subjects (51.2%) experienced a decrease in FVC%pred and 40 (48.8%) an increase. The median annual FVC%pred change in the whole group was −0.1159% (IQR −1.773–2.144). Among patients who declined, the median decrease in FVC%pred was −1.72% per year (IQR −3.7 to −0.78). Among patients who improved, the median annual increase in FVC%pred was 2.16% (IQR 1.03–6.982). The decrease was significantly more prevalent in patients with the GT/TT genetic variant than in those with the GG variant (Table 2).

The probability of FVC%pred decline according to *MUC5B* status is presented in Figure 1. 

In the multiple logistic regression analysis, the presence of the T allele was a single parameter increasing the odds of having a decline in FVC%pred by more than three times (OR 3.598, 95%CI 1.299–10.87, *p* = 0.0171)—Table 3. 

### 2.3. Treatment Effect Assessment According to MUC5B Genotype

In the treated group of 75 patients, PFTs’ parameters were assessed after the median of 12 (IQR 12–12) months from the treatment initiation. The results are presented in Table 4. 

In the group with the GG genotype, we found a significant improvement in both absolute and %pred FVC, TL,co%pred, 6MWD, and DSP after 12 months of treatment with immunomodulatory drugs. On the other hand, among subjects with the presence of at least one T allele (GT/TT), during the treatment, only FVC%pred improved significantly, the improvement of absolute FVC and TL,co%pred was borderline significant, and 6MWT parameters (distance and DSP) did not change. 

## 3. Discussion

Hypersensitivity pneumonitis comprises 20% of all ILDs presenting with progressive pulmonary fibrosis [25]. The pathogenesis of HP is based on the pathologic immunologic response to inhaled, mostly organic antigens, with granuloma formation [26]. Nevertheless, subsequent evolution to fibrotic lung disease is observed in some patients, markedly worsening their life quality and prognosis. The development of lung fibrosis in HP may depend on several risk factors, among others: the type of antigen and its concentration, type of exposition, and genetic predisposition of the patient [26]. Different antigens may trigger different types of immune response and different outcomes; e.g., avian antigens are suspected to produce more severe and fibrotic disease [27]. On the other hand, cigarette smoking may decrease susceptibility to developing HP [28]. However, in our group, the proportion of smokers was similar to the general Polish population in the years 2009–2019 (31–21%, respectively) [29].

The frequency of the T allele of rs35705950 has been found in around 10% of a healthy population of European ancestry, and was much less common in South Asian, East Asian, and African populations (8, 1, and 0.2%, respectively) [30,31,32]. MUC5B polymorphism was found much more frequently in IPF patients than in healthy population, and was an established risk factor for lung fibrosis development and progression. Therefore, we decided to assess the prevalence and clinical significance of the MUC5B promotor region polymorphism (rs35705950) in consecutive HP patients, diagnosed and monitored in our department. We found a minor allele (T) frequency (MAF) of 17%, with the following distribution of genotypes: GG—in 69.8%, GT—in 27,9%, and TT—in 2.3% of patients. These data are in line with the results of Furusawa et al., who found the MAF of 19% and 37% of the GT/TT genotype in the lung tissue obtained from biopsies or explanted lungs of the patients with HP and confirmed the strong association between MUC5B polymorphism and disease development [33]. However, Ley et al. showed a bigger MAF of 24–32% in two cohorts of more than 200 American HP patients with a prevalence of GT/TT genotypes of 41–51% [21]. He also found a correlation between MUC5B gene promoter region polymorphism and the extent of fibrosis in the CT scan [21]. Both of the above-described cohorts were older, contained more ever-smokers, and had worse functional parameters compared to our population, which might have an impact on the higher prevalence of GT/TT genotypes in the above-mentioned study. The other reason for lower GT/TT genotypes in our population compared to the study of Ley et al. may be caused by the fact that we assessed MUC5B promoter polymorphism at a certain time point. Thus, we could not include the patients presenting the worst clinical profile, who died before the date when MUC5B was tested. 

We found significantly lower baseline FVC (absolute and %pred) in patients with the GT/TT genotype compared to those diagnosed with the GG genotype, which could suggest more prevalent fibrotic HP among patients with the GT/TT genotype. Although we found more patients with fibrotic lung disease among those with the GT/TT genotype compared to the GG genotype (82.8% and 63%, respectively), these differences were not significant. This may be due to the small number of non-fibrotic HP patients, who comprised 34% of our cohort only. No association of MUC5B polymorphism with the presence of honeycombing was noted in our cohort, similar to the findings of Ley et al. [21]. The data on FVC in patients with HP in relation to the MUC5B genotype had not been analyzed in the literature yet. 

Furthermore, we analyzed the FVC trajectories in HP patients, depending on MUC5B gene polymorphism. In the entire cohort, the annual FVC%pred decrease and increase trajectories were observed in an equal proportion of patients—51% and 49%, respectively. When analyzing the GG and GT/TT groups separately, significantly more subjects experienced FVC%pred decrease among those with at least one T allele, compared to those expressing the GG genotype (69% vs. 43%, *p* = 0.03). 

Multivariate analysis revealed that the GT/TT genotype was the single, independent predictor of disease progression, increasing the odds of annual FVC%pred decline by 3.6 times. No influence of age, baseline FVC%pred, and the use of treatment on clinical HP course was noted. The association between the annual change of FVC%pred and the presence of the T allele had not been reported yet. This finding stays in opposition to the data obtained in IPF patients, where MUC5B polymorphism, although increasing the risk of disease development, predicts a milder disease course and better survival [32,34]. Only in one study by Jiang et al. was the presence of the T allele of the MUC5B gene combined with decreased pulmonary function tests (FVC and TL,co) and worse survival in IPF patients [35].

We also analyzed the subgroup of 75 treated patients separately. All subjects received systemic corticosteroids, and 43% additionally received an immunosuppressive agent (azathioprine). In the GG genotype group, all PFT and 6MWT parameters improved significantly after a year of treatment, whereas, in the group with the T allele, only FVC (absolute and %pred) significantly improved, while the non-significant improvement of TL,co%pred and stabilization of 6MWT parameters was noted. Thus, we suppose that HP patients with the GT and TT genotypes may express a worse response to immunosuppressive therapy, compared to the GG genotypes. The association between the MUC5B minor allele and the effect of immunomodulatory treatment in HP patients had not been studied before. 

Nintedanib was registered for progressive fibrotic HP patients in 2019. In our country, antifibrotic treatment is used in fibrotic HP only after first-line (i.e., immunomodulatory) treatment failure. We observed numerically a more frequent use of nintedanib in patients with the GT/TT genotype compared to the GG genotype (22 vs. 8%), which was borderline significant, probably due to the small number of patients who qualified for such therapy. As the patients with the T allele tend to respond to immunomodulatory therapy less spectacularly, they need perhaps an earlier consideration of antifibrotic treatment. This hypothesis should be the subject of large prospective clinical trials in the future.

### Study Limitations

Our study has limitations.

First, it was a single-center study with a limited number of patients included. Second, we included both incident and prevalent cases and performed the genetic testing at a particular time point during their follow-up. This may produce immortal time bias, as we do not know the prevalence of MUC5B polymorphism in patients who died before the study was initiated. Additional selection bias is related to the limitations in transportation during the COVID-19 pandemic—patients living a long distance away from our center could not come for a visit and, therefore, could be omitted. For the same reason, patients in worse physical condition could not be included. Third, the proportion of patients with non-fibrotic HP and who were not treated was low, so the differences regarding the influence of MUC5B polymorphism on fibrosis development and treatment response might be under-recognized. Fourth, we tested our cohort for only one genetic variant, which is the strongest but not the single predictor of disease development. Finally, we did not include the healthy control group to assess the baseline frequency of the T allele of the MUC5B gene, which was due to the COVID-19 pandemic and limited access to medical resources. 

## 4. Material and Methods

### 4.1. Regulatory Board Approval

The study was approved by the Institutional Ethics Committee of the National Research Institute of Tuberculosis and Lung Diseases (Approval No. KB-19/2019, date of approval 27 February 2019). All patients gave written informed consent to the genetic testing of the blood samples. 

### 4.2. Study Group

The study group of our retrospective case-control study consisted of 86 consecutive incident and prevalent HP patients admitted to the single pulmonary department from April 2019 to December 2022. All subjects received the diagnosis of HP based on the criteria proposed by Vasakova et al. [1], which was then revised according to the newest ATS/JRS/ALAT diagnostic guidelines [2]. The standard workup used in our center was described in the previous publications [36,37].

Patients’ demographic data, smoking status, and computed tomography (CT) results were collected. Pulmonary function tests (PFTs) and 6 min walking test (6MWT) were performed at baseline, before treatment, and 12 months after treatment implementation. For the purpose of the annual forced vital capacity (FVC) change assessment, historical FVC data from patients’ files were extracted.

### 4.3. Study Procedures

For genetic testing, a single sample (3 whole blood tubes of 9 mL each, S-Monovette^®^ K3 EDTA 9 mL, SARSTEAD, Nümbrecht, Germany) of peripheral blood was drawn from each patient during venipuncture. Samples were adequately labeled with the unique patient’s number to maintain anonymization. 

DNA isolation was carried out using commercial kits (QIAamp DNA Blood Mini Kit, Qiagen, Hilden, Germany), that allow us to obtain a sufficient amount of high-quality DNA material.

Isolated DNA was used in MUC5B genotyping reactions. The presence of a single nucleotide polymorphism (SNP) in the promoter region of the MUC5B gene (rs35705950) was tested with the usage of a commercially available TaqMan SNP Genotyping Assay (C_1582254_20). To detect SNP target, two allele-specific TaqMan minor groove binder (MGB) probes differently labeled with fluorescent dyes (FAM and VIC) with a fluorescence quencher, as well as primer pair uniquely aligned with the studied genome region, was used (the context sequence [VIC/FAM]: CCTTCCTTTATCTTCTGTTTTCAGC[G/T]CCTTCAACTGTGAAGAGGTGAACTC). During reaction nuclease activity of Taq, DNA polymerase was used for cleaving the reporter dye (FAM or VIC) from an MGB probe hybridized to the DNA strand. When separated from the quencher, the reporter dye fluoresced. Reactions were performed on 96-well plates in the presence of negative and positive controls on Applied Biosystems 7500 Fast Dx Real-Time PCR Instrument.

Minor allele frequency (MAF) was calculated by dividing the number of minor alleles (T) by the number of all alleles in the study population (G, T); e.g., if, in the group of 10 people, 5 had genotype GG, 3 had genotype GT, and 2 had genotype TT, the whole number of the allele was 20, the number of GG allele was 5 × 2 + 3 × 1 = 13, the number of T allele was 3 × 1 + 2 × 2 = 7, and MAF was 7/20 = 0.35 (35%). 

Pulmonary function tests (spirometry and diffusion lung capacity) were performed as routine procedures using Master Screen Body/Diffusion (Jaeger, Wuppertal, Germany, 2002; CareFusion, Wurmlingen, Germany, 2017). The results of measurements were presented as absolute values, and the percent of predicted values using the European Respiratory Society (ERS) reference equations. The transfer factor of the lungs for carbon monoxide (TL,co) was measured with a single-breath method using helium or methane (CH_4_) gas as the marker, as described previously [37]. The results were presented as a percentage of predicted values with a correction to hemoglobin concentration [38]. 

Then, 6MWT was performed as a routine procedure in every patient who was able to walk on a flat 30 m long corridor, with baseline and sixth-minute room air oxygen saturation measured with a finger pulse oximeter, according to ATS and Polish Respiratory Society guidelines as described previously [39,40,41].

All chest CT scans were performed in our center (GE Healthcare Revolution GSI scanner) in the course of the diagnostic process and were analyzed by two radiologists with expertise in ILDs. The results were presented according to the internal references, including the morphological description of the visible abnormalities (particularly the location and distribution of the changes, presence of nodules, ground glass opacities, air trapping—if applicable—in expiration scans, three-density pattern, reticulations, traction bronchiectases, and honeycombing) and the final conclusion of typical HP, compatible with HP or indeterminate for HP. In case of discrepancies between radiologists regarding the final conclusion, a multidisciplinary discussion was conducted, and the result was established. Therefore, the results of chest CT scans were used as source data without re-evaluation. 

The treatment initiation was based on a case-by-case clinical assessment. The main indications for treatment were low baseline PFT results and/or respiratory failure, and lack of improvement or worsening in PFTs and chest X-ray despite the antigen avoidance trial. The patients were treated either with corticosteroids alone or in combination with azathioprine. Those who had worsening clinical, functional, and radiological features despite immunomodulatory treatment could receive an antifibrotic drug (i.e., nintedanib). Patients with normal or slightly impaired PFTs and stable chest X-rays were not treated.

The observation period was censored on 31st December 2022. The disease duration time was calculated from the diagnosis to the censoring date, lung transplantation, or death. All patients had at least a 12-month-long observation. 

### 4.4. Assessment of the Disease Course

Baseline PFT parameters and subsequent FVC and TL,co measurements were taken into account. In patients who had at least two measurements of FVC during clinical observation, the annual change in FVC%pred was calculated as follows: $\mathrm{annual}\;\mathrm{FVC}\%\mathrm{pred}\;\mathrm{change}=\frac{\mathrm{last}\;\mathrm{available}\;\mathrm{FVC}\%\mathrm{pred}-\mathrm{baseline}\;\mathrm{FVC}\%\mathrm{pred}}{\mathrm{number}\;\mathrm{of}\;\mathrm{years}\;\mathrm{of}\;\mathrm{observation}}$. In patients who received the treatment, PFTs parameters were evaluated before treatment initiation and after 12 months of treatment. Numerical changes in each parameter were taken into consideration. 

### 4.5. Statistical Analysis

The statistical analysis was performed using GraphPad Prism (GraphPad Software, 9.4.1 (458) on 18 July 2022, LCC, San Diego, CA, USA). The values were presented as means ± SD or medians and ranges. Between-group comparison for continuous variables in two groups was assessed with Student’s *t*-test, Fisher’s exact test, Mann–Whitney test, or Wilcoxon test, where appropriate. *p* values of <0.05 were considered statistically significant. Multiple variables were analyzed using multiple logistic regression analysis. Minor allele frequency (MAF) was calculated by dividing the number of the minor allele (T) by the number of all alleles.

## 5. Conclusions

The MUC5B minor allele (T) was present in around 30% of patients with HP, who were diagnosed and observed in our department. We found significantly lower baseline FVC%pred and a greater decline in FVC in the course of the disease in patients with the GT/TT genotype. Those patients also tended to have more fibrotic changes in HRCT and were more frequently treated with antifibrotics, but the differences were not significant. If treated with immunomodulatory drugs, patients with the presence of the T allele experienced less pronounced improvements in the functional parameters.

Therefore, according to our results, the MUC 5B GT/TT genotype in HP patients may be combined with lower FVC%pred at diagnosis, a worse response to immunomodulatory treatment, and the need for an earlier implementation of antifibrotic treatment. 

Nevertheless, more prospective data and multi-center studies are needed to confirm those findings. 

## Figures and Tables

**Figure 1 ijms-24-10748-f001:**
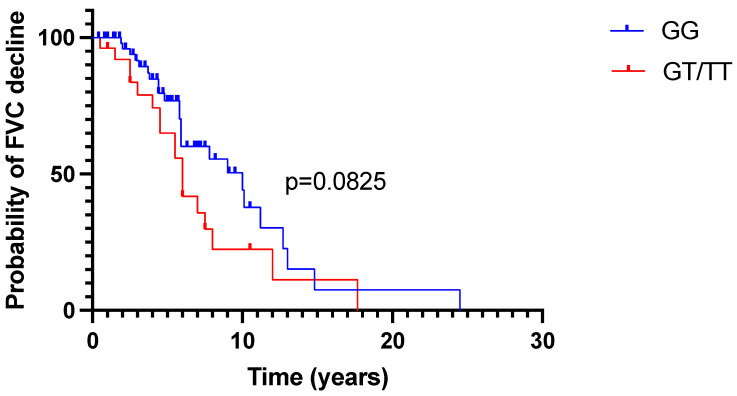
Probability of FVC%pred decline during observation according to *MUC5B* status. FVC—forced vital capacity, GG—GG genotype, GT/TT—GT/TT genotype.

**Table 1 ijms-24-10748-t001:** Baseline characteristics of the study group according to *MUC5B* genotype.

Variable		Whole Group *n* = 86	*MUC5B* GG Genotype *n* = 60	*MUC5B* GT/TT Genotype *n* = 26	*p* Value (GG vs. GT/TT)
Age at diagnosis (y), median (range)		58 (20–75)	59 (23–75)	53 (20–74)	0.3454
Follow-up time (mth), median (range)		62.4 (2.2–293.9)	54.8 (2.2–293.9)	67.7 (4.9–205.8)	0.2794
Males, *n* (%)		38 (44)	29 (48.3)	9 (34.6)	0.3445
Ever smokers, *n* (%)		27 (31.5)	20 (33.3)	7 (26.9)	0.6207
FVC (L), median (range)		2.6 (1.52–5.68)	2.97 (1.52–5.68)	2.48 (1.53–5.45)	0.0610
FVC (%pred), mean (±SD)		79.1 (±19.3)	82.18 (±19.04)	72.28 (±18.29)	0.0315
TL,co (%pred), median (range)		57 (28–114)	54.5 (28–113)	59 (32–114)	0.8695
6MWD (m), mean (±SD)		509.5 (±101.4)	505.6 (±113.5)	518.0 (±69.96)	0.6180
DSP, mean (±SD)		459.4 (±101.0)	457.7 (±113.6)	463.1 (±66.16)	0.6848
GAP stage (*n* = 83)	1	65 (78.3)	45 (77.6)	20 (83.3)	0.7748
	2	15 (18.1)	11 (19)	3 (12.5)	
	3	2 (3.6)	2 (3.4)	1 (4.2)	
Fibrotic HP *n* (%)		57 (66.3)	36 (63.2)	21 (82.8)	0.0828
Immunomodulatory treatment *n* (%)		75 (87.2)	52 (86.7)	23 (88.5)	0.9999
Antifibrotic treatment *n* (%)		11 (12.8)	5 (8.2)	6 (22.2)	0.0853

FVC—forced vital capacity; TL,co—transfer factor of the lungs for carbon monoxide; 6MWD—6 min walking distance; DSP—distance–saturation product; GAP—gender–age–physiology index; HP—hypersensitivity pneumonitis.

**Table 2 ijms-24-10748-t002:** Annual FVC%pred trajectory according to *MUC5B* genotype.

FVC%Pred Trajectory	MUC5B GG Genotype, *n* (%)	MUC5B GT/TT Genotype, *n* (%)	Total, *n* (%)	*p* Value
FVC%pred increase	32 (57.2)	8 (30.8)	40 (48.8)	0.0336
FVC%pred decrease	24 (42.8)	18 (69.2)	42 (51.2)	
Total, *n* (%)	56 (68.3)	26 (31.7)	82 (100)	

FVC%pred—forced vital capacity % predicted.

**Table 3 ijms-24-10748-t003:** Factors predicting the decline in FVC%pred with time (multiple logistic regression analysis).

Variable	OR	95%CI	*p* Value
T allele (YES)	3.598	1.299–10.87	0.0171
Treatment (NO)	0.2391	0.0397–1.130	0.0865
Age at diagnosis	1.003	0.9681–1.039	0.8867
Baseline FVC%pred	1.015	0.9885–1.044	0.2713

FVC%pred—forced vital capacity % predicted, OR—odds ratio, 95% CI—95% confidence interval.

**Table 4 ijms-24-10748-t004:** Changes in the functional parameters after 12 months of immunomodulatory treatment according to the genotype (Wilcoxon test).

Parameter	GG Genotype		*p* Value (GG Pre vs. GG Post)	GT/TT Genotype		*p* Value (GT/TT Pre vs. GT/TT Post)
	Pretreatment	Posttreatment		Pretreatment	Posttreatment	
FVC (l), median (range)	2.655 (1.52–5.68) *	2.995 (1.51–5.03) *	0.0014	2.12 (1.23–5.45) ^@^	2.21 (1.3–5.83) ^@^	0.0671
FVC (%), mean (±SD)	78.87(± 19.93) ^#^	85.67(± 21.94) ^#^	0.0015	67.19 (±16.56) ^&^	72.46 (±19.08) ^&^	0.0082
TL,co (%), median (range)	51(28–113) ^#^	58.5 (33–114) ^#^	0.0004	55.5 (31–114) ^@^	59 (31–119) ^@^	0.0586
6MWD, mean (±SD)	513 (±104.2) ^$^	542.2 (±98.94) ^$^	0.0106	524.8 (±78.95) ^@^	529.1 (±92.20) ^@^	0.8195
DSP, mean (±SD)	457.5 (±106.2) ^$^	490 (±104.6) ^$^	0.0029	465.5 (±79.57) ^@^	465.6 (±101.7) ^@^	0.8117

FVC—forced vital capacity, TL,co—transfer factor of the lungs for carbon monoxide, 6MWD—6 min walking distance, DSP—distance–saturation product, *—*n* = 48, ^#^-*n* = 49, ^$^—*n* = 46, ^@^—*n* = 22, ^&^—*n* = 23.

## Data Availability

Data are available from the corresponding author upon request.

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
