# Peer review of "The Presence of T Allele (rs35705950) of the MUC5B Gene Predicts Lower Baseline Forced Vital Capacity and Its Subsequent Decline in Patients with Hypersensitivity Pneumonitis"

_ijms, 2023, doi:10.3390/ijms241310748_

Round 1
Reviewer 1 Report
In this study, the authors establish the frequency of minor allele of the MUC5B gene in the population of patients with hypersensitivity pneumonitis and find the relationship between the MUC5B promoter region polymorphism and the development of lung fibrosis, the severity of the disease course, and the response to the treatment in patients with hypersensitivity pneumonitis
- When abbreviations are used, spell out the full word at first mention in the text followed by the abbreviation in the parentheses. Thereafter, use the abbreviation throughout.
- Tables should be clearer because the reader should have difficulty reading the data horizontally. Furthermore, the legends should be enriched
- The authors should enrich the paragraph on the conclusions
- In section 3.3 the authors should better explain the results
Reviewer 2 Report
Comments to the Author
The authors showed frequency ofMUC5B polymorphism and association among that, FVC decline and efficacy of treatment. The present study has novelty and clinical significance. The resolution of the following issues for acceptance of International Journal of Molecular Science.
Major comment
1. Please show the previous published paper that served as a reference for the evaluation method of the disease course based on the FVC trajectory. I consider that the cut-off value for FVC decline should be 5% or 10%, which is used as a criterion for PPF and were correlated with mortality in fibrotic ILD.
2. Please show the method of consensus when there is a discrepancy between the evaluations of the two radiologist who evaluated CT findings.
Round 2
Reviewer 2 Report
Autor responsed to my query correctly.